# A Unified Approach to Local Quantum Uncertainty and Interferometric Power by Metric Adjusted Skew Information

**DOI:** 10.3390/e23030263

**Published:** 2021-02-24

**Authors:** Paolo Gibilisco, Davide Girolami, Frank Hansen

**Affiliations:** 1Department of Economics and Finance, University of Rome “Tor Vergata”, Via Columbia 2, 00133 Rome, Italy; 2Dipartimento di Scienza Applicata e Tecnologia, Politecnico di Torino, Corso Duca degli Abruzzi 24, 10129 Torino, Italy; davide.girolami@polito.it; 3Department of Mathematical Sciences, University of Copenaghen, Universitetsparken 5, DK-2100 Copenhagen, Denmark; frank.hansen@math.ku.dk

**Keywords:** fisher information, operator monotone functions, matrix means, quantum Fisher information, metric adjusted skew information, local quantum uncertainty, interferometric power

## Abstract

Local quantum uncertainty and interferometric power were introduced by Girolami et al. as geometric quantifiers of quantum correlations. The aim of the present paper is to discuss their properties in a unified manner by means of the metric adjusted skew information defined by Hansen.

## 1. Introduction

One of the key traits of many-body quantum systems is that the full knowledge of their global configurations does not imply full knowledge of their constituents. The impossibility of reconstructing the local wave functions $|\psi_{1}\rangle ,\;|\psi_{2}\rangle$ (pure states) of two interacting quantum particles from the wave function of the whole system, $|\psi_{12}\rangle \neq |\psi_{1}\rangle ⊗|\psi_{2}\rangle$, is due to the existence of entanglement [1]. Investigating open quantum systems, whose (mixed) states are described by density matrices $\rho_{12}=\sum_{i}p_{i}|\psi_{i}\rangle_{12}\langle \psi_{i}|$, revealed that the boundary between the classical and quantum worlds is more blurred than we thought. There exists a genuinely quantum kind of correlation, quantum discord, which manifests even in the absence of entanglement, i.e., in separable density matrices $\rho_{12}=\sum_{i}p_{i}\rho_{1,i}⊗\rho_{2,i}$ [2,3]. This discovery triggered theoretical and experimental studies to understand the physical meaning of quantum discord, and the potential use of it as a resource for quantum technologies [4]. Relying on the known interplay between the geometrical and physical properties of mixed states [5,6], a stream of works employed information geometry techniques to construct quantifiers of quantum discord [7,8,9,10,11,12]. In particular, two of the most popular ones are the Local Quantum Uncertainty (LQU) and the Interferometric Power (IP) [13,14]. A merit of these two measures is that they admit an analytical form for *N* qubit states across the 1vsN−1 qubit partition. They also have a clear-cut physical interpretation. The lack of certainty about quantum measurement outcomes is due to the fact that density matrices are changed by quantum operations. The LQU evaluates the minimum uncertainty about the outcome of a local quantum measurement, when performed on a bipartite system. It has been proven that two-particle density matrices display quantum discord if, and only if, they are not “classical-quantum” states—that is, they are not (a mixture of) eigenvalues of local observables, $\rho_{12}\neq \sum_{i}p_{i}{|i\rangle }_{1}\langle i|⊗\rho_{2,i}$, or $\rho_{12}\neq \sum_{i}p_{i}\rho_{1,i}⊗{|i\rangle }_{2}\langle i|$, in which {|i〉} is an orthonormal basis. Indeed, this is the only case in which one can identify a local measurement that does not change a bipartite quantum state, whose spectral decomposition reads $A_{1}=\sum_{i}\lambda_{i}{|i\rangle }_{1}\langle i|$, or $A_{2}=\sum_{i}\lambda_{i}{|i\rangle }_{2}\langle i|$. The LQU was built as the minimum of the Wigner–Yanase skew information, a well-known information geometry measure [15], between a density matrix and a finite-dimensional observable (Hermitian operator). It quantifies how much a density matrix $\rho_{12}$ is different from being a zero-discord state. The IP was concocted by following a similar line of thinking. Quantum discord implies a non-classical sensitivity to local perturbations. This feature of quantum particles, while apparently a limitation, translates into an advantage in the context of quantum metrology [16]. It has been theoretically proven and experimentally demonstrated that quantum systems sharing quantum discord are more sensitive probes for interferometric phase estimation. The merit of such measurement protocols is the quantum Fisher information of the state under scrutiny with respect to a local Hamiltonian (in Information Geometry, the QFI is known as the SLD or Bures–Uhlmann metric). The latter generates a unitary evolution that imprints information about a physical parameter on the quantum probe. The IP is the minimum quantum Fisher information of all the possible local Hamiltonians, which is zero if, and only if, the probe states are classically correlated.

Here, we polish and extend the mathematical formalization of information-geometric quantum correlation measures. We build a class of parent quantities of the LQU (and consequently of the IP) in terms of the the metric adjusted skew informations [17]. In Section 2 and Section 3, we review the definition and main properties of operator means. In Section 4, Section 5 and Section 6, we discuss information-geometric quantities that capture complementarity between quantum states and observables. In particular, we focus on the quantum *f*-covariances and the quantum Fisher information. They quantify the inherent uncertainty about quantum measurement outcomes. After recalling the definition of metric adjusted skew information (Section 7), we build a new quantum discord measure, the metric adjusted local quantum uncertainty (*f*-LQU), in Section 8. Finally, we are able to show that LQU and IP are just two particular members of this family, allowing a unified treatment of their fundamental properties.

## 2. Means for Positive Numbers

We use the notation $R_{+}=\left(0,+\infty \right)$.

**Definition** **1.**
*A bivariate mean [18] is a function $m:R_{+}\times R_{+}\to R_{+}$ such that:*
*1*.
m(x,x)=x;
*2*.
m(x,y)=m(y,x);
*3*.
x<y
⇒
x<m(x,y)<y;
*4*.
$x<x^{\prime }$
*and*
$y<y^{\prime }$
⇒
$m\left(x,y\right)<m\left(x^{\prime },y^{\prime }\right);$
*5*.
*m is continuous;*
*6*.
*m is positively homogeneous; that is m(tx,ty)=t·m(x,y) for t>0.*



We use the notation $M_{nu}$ for the set of bivariate means described above.

**Definition** **2.**
*Let $F_{nu}$ denote the class of functions $f:R_{+}\to R_{+}$ such that:*
*1*.
*f is continuous;*
*2*.
*f is monotone increasing;*
*3*.
*f(1)=1;*
*4*.
*$tf\left(t^{-1}\right)=f\left(t\right)$.*



The following result is straightforward.

**Proposition** **1.**
*There is a bijection $f\mapsto m_{f}$ betwen $F_{nu}$ and $M_{nu}$ given by*
$$m_{f}\left(x,y\right)=yf\left(y^{-1}x\right)\;\mathrm{and}\;\mathrm{in}\;\mathrm{reverse}\;f\left(t\right)=m\left(1,t\right)$$
*for positive numbers x,y and t.*


In Table 1, we have some examples of means.

## 3. Means for Positive Operators in the Sense of Kubo-Ando

The celebrated Kubo–Ando theory of operator means [18,19,20] may be viewed as the operator version of the results of Section 2.

**Definition** **3.**
*A bivariate mean m for pairs of positive definite operators is a function*
(A,B)→m(A,B),
*defined in and with values in positive definite operators on a Hilbert space that satisfies mutatis mutandis conditions (1) to (5) in Definition 1. In addition, the transformer inequality*
$$Cm\left(A,B\right)C^{*}\leq m\left(CAC^{*},CBC^{*}\right),$$
*should also hold for positive definite A,B, and arbitrary invertible C.*


Note that the transformer inequality replaces condition (6) in Definition 1. We denote the set of matrix means by $M_{op}$.

**Example** **1.**
*The arithmetic, geometric and harmonic operator means are defined, respectively, by setting*
$$\begin{array}{ccc} A\nabla B & = & \frac{1}{2}\left(A+B\right)\\ A\#B & = & A^{1/2}{\left(A^{-1/2}BA^{-1/2}\right)}^{1/2}A^{1/2}\\ A!B & = & 2{\left(A^{-1}+B^{-1}\right)}^{-1}. \end{array}$$


We recall that a function f:(0,∞)→R is said to be *operator monotone (increasing)* if
A≤B⇒f(A)≤f(B)
for positive definite matrices of arbitrary order. It then follows that the inequality also holds for positive operators on an arbitrary Hilbert space. An operator monotone function *f* is said to be *symmetric* if $f\left(t\right)=tf\left(t^{-1}\right)$ for t>0 and *normalized* if f(1)=1.

**Definition** **4.**
*$F_{op}$ is the class of functions $f:R_{+}\to R_{+}$ such that:*
*1*.
*f is operator monotone increasing;*
*2*.
$tf\left(t^{-1}\right)=f\left(t\right)\;t>0;$
*3*.
f(1)=1.



**Remark** **1.**
*In $F_{op}$ the functions*
$$\frac{1+x}{2}\;\mathrm{and}\;\frac{2x}{1+x}$$
*are, respectively, the biggest and the smallest element.*


The fundamental result, due to Kubo and Ando, is the following.

**Theorem** **1.**
*There is a bijection $f\mapsto m_{f}$ between $M_{op}$ and $F_{op}$ given by the formula*
$$m_{f}\left(A,B\right)=A^{1/2}f\left(A^{-1/2}BA^{-1/2}\right)A^{1/2}.$$


Following Remark 1, we have the inequalities
$$\frac{2}{A^{-1}+B^{-1}}\leq m_{f}\left(A,B\right)\leq \frac{A+B}{2}$$
which are valid for any $f\in F_{op}\;,$ cf. [20] (Theorem 4.5).

**Remark** **2.**
*The functions in $F_{op}$ are (operator) concave, which makes the operator case quite different from the numerical (commutative) case. For example, there are convex functions in $F_{nu}\;,$ cf. [21].*


If ρ is a density matrix (a quantum state) and *A* is a self-adjoint matrix (a quantum observable), then the expectation of *A* in the state ρ is defined by setting
$$E_{\rho }\left(A\right)=\mathrm{Tr}\;\;\left(\rho A\right).$$

## 4. Quantum ***f***-Covariance

The notion of quantum *f*-covariance was introduced by Petz; see [22,23]. Any Kubo–Ando function $m_{f}\left(x,y\right)=yf\left(y^{-1}x\right)$ for x,y>0 has a continuous extension to [0,+∞)×[0,+∞), given by
$$m_{f}\left(0,y\right)=f\left(0\right)y,\;m_{f}\left(x,0\right)=f\left(0\right)x,\;m_{f}\left(0,0\right)=0,\;x,y>0.$$The operator $m_{f}\left(L_{\rho },R_{\rho }\right)$ is well-defined by the spectral theorem for any state; see [24] (Proposition 11.1 page 11). To self-adjoint A, we set $A_{0}=A-\left(\mathrm{Tr}\;\;\rho A\right)I,$ where *I* is the identity operator. Note that
$$\mathrm{Tr}\;\;\rho A_{0}=\mathrm{Tr}\;\;\rho A-\left(\mathrm{Tr}\;\;\rho A\right)\mathrm{Tr}\;\;\rho =0,$$
if ρ is a state.

**Definition** **5.**
*Given a state ρ, a function $f\in F_{op}$ and self-adjoint A,B, we define the quantum f-covariance by setting*
$${\mathrm{Cov}}_{\rho }^{f}\left(A,B\right)=\mathrm{Tr}\;\;B_{0}\;m_{f}\left(L_{\rho },R_{\rho }\right)A_{0}$$
*and the corresponding quantum f-variance by ${\mathrm{Var}}_{\rho }^{f}\left(A\right)={\mathrm{Cov}}_{\rho }^{f}\left(A,A\right).$*


The *f*-covariance is a positive semi-definite sesquilinear form and
(1)$$f\leq g\;\Rightarrow \;{\mathrm{Var}}_{\rho }^{f}\left(A\right)\leq {\mathrm{Var}}_{\rho }^{g}\left(A\right).$$

Note that for the standard covariance, we have ${\mathrm{Cov}}_{\rho }\left(A,B\right)={\mathrm{Cov}}_{\rho }^{SLD}\left(A,B\right),$ where the SLD or Bures–Uhlmann metric is the one associated with the function (1+x)/2 (see the end of Section 5).

## 5. Quantum Fisher Information

The theory of quantum Fisher information is due to Petz, and we recall the basic results. If N is a differentiable manifold, we denote by $T_{\rho }N$ the tangent space to N at the point ρ∈N. Let $M_{n}$ (resp. $M_{n,sa}$) be the set of all complex n×n matrices (respectively, of all complex self-adjoint n×n matrices). The set of faithful states is defined as
$$D_{n}^{1}=\left\{\rho \in M_{n,sa}|\mathrm{Tr}\;\rho =1,\;\;\rho >0\right\}.$$

Recall that there exists a natural identification of $T_{\rho }D_{n}^{1}$ with the space of self-adjoint traceless matrices; namely, for any $\rho \in D_{n}^{1}$
$$T_{\rho }D_{n}^{1}=\left\{A\in M_{n}|A=A^{*}\;,\;\mathrm{Tr}\;A=0\right\}.$$

A stochastic map is a completely positive and trace-preserving operator $T:M_{n}\to M_{m}\;.$ A *monotone metric* is a family of Riemannian metrics $g=\{g^{n}\}$ on $\left\{D_{n}^{1}\right\}$, n∈N, such that the inequality
$$g_{T(\rho )}^{m}\left(TX,TX\right)\leq g_{\rho }^{n}\left(X,X\right)$$
holds for every stochastic map $T:M_{n}\to M_{m}$, every faithful state $\rho \in D_{n}^{1},$ and every $X\in T_{\rho }D_{n}^{1}\;.$ Usually, monotone metrics are normalized in such a way that [A,ρ]=0 implies $g_{\rho }\left(A,A\right)=\mathrm{Tr}\left(\rho^{-1}A^{2}\right)$. A monotone metric is also called (an example of) *quantum Fisher information* (QFI). This notation is inspired by Chentsov’s uniqueness theorem for commutative monotone metrics [25].

Define $L_{\rho }\left(A\right)=\rho A$ and $R_{\rho }\left(A\right)=A\rho$, and observe that $L_{\rho }$ and $R_{\rho }$ are commuting positive superoperators on $M_{n}.$ For any $f\in F_{op}$, one may also define the positive superoperator $m_{f}\left(L_{\rho },R_{\rho }\right)$. The fundamental theorem of monotone metrics may be stated in the following way; see [26].

**Theorem** **2.**
*There exists a bijective correspondence between symmetric monotone metrics (sometimes called quantum Fisher informations) on $D_{n}^{1}$ and functions $f\in F_{op\;.}$ The correspondence is given by the formula*
$${\langle A,B\rangle }_{\rho ,f}=\mathrm{Tr}\;\;A\;m_{f}{\left(L_{\rho },R_{\rho }\right)}^{-1}\left(B\right)$$
*for positive definite matrices A and B.*


**Remark** **3.**
*The reader should be aware that, in the physics literature, the name Quantum Fisher Information is used to denote a specific monotone metric, namely the one associated to the function f(x)=(1+x)/2, which is also known as the Symmetric Logarithmic Derivative metric or the Bures–Uhlmann metric.*


## 6. The $f\to \tilde{f}$ correspondence

We introduce a technical tool which is useful for establishing some fundamental relations between quantum covariance, quantum Fisher information and the metric adjusted skew information.

**Definition** **6.**
*For $f\in F_{op}$ we define $f\left(0\right)=\lim_{x\to 0}f\left(x\right).$ It is meaningful since f is increasing. We say that a function $f\in F_{op}$ is regular if f(0)≠0, and non-regular if f(0)=0, cf. [17,27].*


**Definition** **7.**
*A quantum Fisher information is extendable if its radial limit exists, and it is a Riemannian metric on the real projective space generated by the pure states.*


For the definition of the radial limit, see [27], where the following fundamental result is proved.

**Theorem** **3.**
*An operator monotone function $f\in F_{op}$ is regular, if and only if ${\langle \cdot ,\cdot \rangle }_{\rho ,f}$ is extendable.*


**Remark** **4.**
*The reader should be aware that there is no negative connotation associated with the qualification “non-regular”. For example, a very important quantum Fisher information in quantum physics [28], namely the Kubo–Mori metric, generated by the function f(x)=(x−1)/logx, is non-regular.*

*We introduce the sets of regular and non-regular functions*
$$F_{op}^{\;r}:=\left\{f\in F_{op}|f\left(0\right)\neq 0\right\}\;\mathrm{and}\;F_{op}^{\;n}:=\left\{f\in F_{op}|f\left(0\right)=0\right\}.$$
*Trivially, $F_{op}=F_{op}^{\;r}\cup F_{op}^{\;n}$.*


**Definition** **8.**
*We introduce to a function $f\in F_{op}^{\;r}$, the transform $\tilde{f},$ given by*
$$\tilde{f}\left(x\right)=\frac{1}{2}\left[\left(x+1\right)-{\left(x-1\right)}^{2}\frac{f(0)}{f(x)}\right]\;x>0.$$

*We may also write $\tilde{f}=G\left(f\right),$ cf. [19,24].*


The following result is taken from [19] (Theorem 5.1).

**Theorem** **4.**
*The correspondence $f\to \tilde{f}$ is a bijection between $F_{op}^{\;r}$ and $F_{op}^{\;n}\;.$*


In Table 2, we have some examples (where 0<β<1).

**Proposition** **2.**
*If ρ is a pure state, then ${\mathrm{Var}}_{\rho }^{f}\left(A\right)=2\;m_{f}\left(1,0\right)\cdot {\mathrm{Var}}_{\rho }\left(A\right),$ cf. [29].*


**Corollary** **1.**
*If ρ is a pure state and f is non-regular, then ${\mathrm{Var}}_{\rho }^{f}\left(A\right)=0.$*


**Proof.** If *f* is non-regular $m_{f}\left(1,0\right)=0.$ □

## 7. Metric Adjusted Skew Information

By using the general form of the quantum Fisher information, it is possible to greatly generalize the Wigner–Yanase information measure. To a function $f\in F_{op},$ the so-called Morosova function $c_{f}\left(x,y\right)$ is defined by setting
(2)$$c_{f}\left(x,y\right)=\frac{1}{yf(xy^{-1})}=m_{f}{\left(x,y\right)}^{-1}\;x,y>0.$$

The corresponding monotone symmetric metric $K_{\rho }$ is given by
(3)$$K_{\rho }^{f}\left(A,B\right)=\mathrm{Tr}\;\;A^{*}c_{f}\left(L_{\rho },R_{\rho }\right)B,$$
where $L_{\rho }$ and $R_{\rho }$ denote left and right multiplication with ρ. Note that $K_{\rho }^{f}\left(A\right)$ is increasing in $c_{f}$, and thus decreasing in f. Furthermore, if *f* is regular, the notion of metric adjusted skew information [17] (Definition 1.2) is defined by setting
(4)$$I_{\rho }^{f}\left(A\right)=I^{f}\left(\rho ,A\right)=\frac{f(0)}{2}K_{\rho }^{f}\left(i\left[\rho ,A^{*}\right],i\left[\rho ,A\right]\right),$$
where ρ>0. We use the second notation, $I^{f}\left(\rho ,A\right),$ when the expression of the state takes up too much space. We also tacitly extended the metric adjusted skew information to arbitrary (not necessarily self-adjoint) operators A. It is convex [17] (Theorem 3.7) in the state variable ρ, and
(5)$$0\leq I_{\rho }^{f}\left(A\right)\leq {\mathrm{Var}}_{\rho }\left(A\right)$$
with equality if ρ is pure [17] (Theorem 3.8); see the summary with interpretations in [30] (Theorem 1.2). Furthermore, the notion of unbounded metric adjusted skew information for non-regular functions in $F_{op}$ is introduced in [30] (Theorem 5.1). For a regular function $f\in F_{op},$ the metric adjusted skew information may be written as
$$I_{\rho }^{f}\left(A\right)=\mathrm{Tr}\;\;\rho A^{2}-\mathrm{Tr}\;\;A\;m_{\tilde{f}}\left(L_{\rho },R_{\rho }\right)A,$$
cf. [31] (Equation (Equation 7)). We thus obtain that the metric adjusted skew information is decreasing in the transform $\tilde{f}$ for arbitrary self-adjoint A, that is
(6)$$\tilde{f}\leq \tilde{g}\;\Rightarrow \;I_{\rho }^{f}\left(A\right)\geq I_{\rho }^{g}\left(A\right)\;\mathrm{for}\;f,g\in F_{op}^{\;r}\;.$$

Therefore, we have the following result.

**Proposition** **3.**
*Setting $f_{SLD}\left(x\right)=\left(1+x\right)/2$ we obtain ${\tilde{f}}_{SLD}=2x/\left(1+x\right)$ and therefore*
$$I_{\rho }^{g}\left(A\right)\leq I_{\rho }^{f_{SLD}}\left(A\right)\;\;\forall g\in F_{op}^{\;r}\;.$$


We may also introduce the transforms
$$\overset{ˇ}{f}=\frac{f(0)}{f(t)}\;\mathrm{and}\;\overset{ˇ}{c}\left(x,y\right)=y^{-1}\overset{ˇ}{f}\left(xy^{-1}\right)$$
and obtain
$$I_{\rho }^{f}\left(A\right)=\frac{1}{2}\mathrm{Tr}\;\;i\left[\rho ,A^{*}\right]\overset{ˇ}{c}\left(L_{\rho },R_{\rho }\right)i\left[\rho ,A\right],$$
cf. [31] (Equation (Equation 10)). It follows that the metric adjusted skew information is increasing in $\overset{ˇ}{f}$ for arbitrary A. It can be derived from [24] (Proposition 6.3, page 11), that the metric adjusted skew information can be expressed as the difference
$$I_{\rho }^{f}\left(A\right)={\mathrm{Var}}_{\rho }\left(A\right)-{\mathrm{Var}}_{\rho }^{\tilde{f}}\left(A\right)$$
with extension to the sesquilinear form
$$I_{\rho }^{f}\left(A,B\right)={\mathrm{Cov}}_{\rho }\left(A,B\right)-{\mathrm{Cov}}_{\rho }^{\tilde{f}}\left(A,B\right).$$

### 7.1. Information Inequalities

A function $f:R_{+}\to R_{+}$ is in $F_{op}$ if and only if it allows a representation of the form
(7)$$f\left(t\right)=\frac{1+t}{2}\exp\int_{0}^{1}\frac{\left(\lambda^{2}-1\right){\left(1-t\right)}^{2}}{\left(\lambda +t\right)\left(1+\lambda t\right){\left(1+\lambda \right)}^{2}}\;h_{f}\left(\lambda \right)\;\;\mathrm{error}\;\lambda ,$$
where the weight function $h_{f}:\left[0,1\right]\to \left[0,1\right]$ is measurable. The equivalence class containing $h_{f}$ is uniquely determined by f, cf. [31] (Theorem 2.1). This representation gives rise to an order relation on the set $F_{op}\;.$

**Definition** **9.**
*Let $f,g\in F_{op}\;.$ We say that f is majorized by g and write f⪯g, if the function*
$$\varphi \left(t\right)=\frac{t+1}{2}\;\frac{f(t)}{g(t)}\;t>0$$
*is in $F_{op}\;.$*


The partial order relation ⪯ is stronger that the usual order relation ≤, and it renders $\left(F_{op}\;,⪯\right)$ into a lattice with
(8)$$f_{\min}\left(t\right)=\frac{2t}{t+1}\;\mathrm{and}\;f_{\max}\left(t\right)=\frac{t+1}{2}\;$$
as, respectively, the minimal element and maximal element. Furthermore,
(9)$$f⪯g\;\mathrm{if}\;\mathrm{and}\;\mathrm{only}\;\mathrm{if}\;h_{f}\geq h_{g}\;\mathrm{almost}\;\mathrm{everywhere},$$
cf. [31] (Theorem 2.4). The restriction of ⪯ to the regular part of $F_{op}$ induces a partial order relation ⪯ on the set of metric adjusted skew informations.

**Proposition** **4.**
*The restriction of the order relation *⪯* renders the regular part of $F_{op}$ into a lattice. In addition, if one of two functions f,g in $F_{op}$ is non-regular, then the minorant f∧g is also non-regular.*


**Proof.** Take $f\in F_{op}$ with representative function $h_{f},$ as given in (Equation 7). It is easily derived that *f* is regular if and only if the weight function $h_{f}$ satisfies the integrability condition
(10)$$\int_{0}^{1}\frac{h_{f}\left(\lambda \right)}{\lambda }\;\;d\;\lambda <\infty .$$Take regular functions $f,g\in F_{op\;.}$ We know that $\left(F_{op}\;,⪯\right)$ is a lattice [31] (bottom of page 141), and that the representative function in (Equation 7) for the minorant f∧g is given by
$$h_{f\wedge g}=\max\left\{h_{f},h_{g}\right\}\leq h_{f}+h_{g}\;.$$The inequality above shows that the weight function $h_{f\wedge g}$ also satisfies the integrability condition (Equation 10), which implies that f∧g is regular. Since
$$h_{f\vee g}=\min\left\{h_{f},h_{g}\right\}\leq h_{f}$$
it also follows that the majorant is regular. We now take functions $f,g\in F_{op}$ with representative functions $h_{f}$ and $h_{g}$ and assume that *f* is non-regular. Since
$$h_{f\wedge g}=\max\left\{h_{f},h_{g}\right\}\;\mathrm{and}\;\mathrm{thus}\;h_{f}\leq h_{f\wedge g}$$
we obtain that also the minorant f∧g is non-regular. □

### 7.2. The Wigner–Yanase–Dyson Skew Informations

The Wigner–Yanase–Dyson skew information (with parameter p) is defined by setting
$$I_{p}\left(\rho ,A\right)=-\frac{1}{2}\mathrm{Tr}\;\;[\rho^{p},A[\left[\rho^{1-p},A\right],\;0<p<1.$$This is an example of a metric adjusted skew information and reduces to the Wigner–Yanase skew information for p=1/2. The representing function $f_{p}$ in $F_{op}^{\;r}$ of $I_{p}\left(\rho ,A\right)$ is given by
$$f_{p}\left(t\right)=p\left(1-p\right)\cdot \frac{{\left(t-1\right)}^{2}}{\left(t^{p}-1\right)\left(t^{1-p}-1\right)}\;0<p<1,$$
that is, $I_{p}\left(\rho ,A\right)=I_{\rho }^{f_{p}}\left(A\right).$ The weight-functions $h_{p}\left(\lambda \right)$ in Equation (Equation 7) corresponding to the representing functions $f_{p},$ are given by
$$h_{p}\left(\lambda \right)=\frac{1}{\pi }\arctan\frac{(\lambda^{p}+\lambda^{1-p})\sin p\pi }{1-\lambda -(\lambda^{p}-\lambda^{1-p})\cos p\pi }\;0<\lambda <1.$$It is non-trivial that the Wigner–Yanase–Dyson skew information $I_{p}\left(\rho ,A\right)$ is increasing in the parameter *p* for 0<p≤1/2 and decreasing in *p* for 1/2≤p<1 with respect to the order relation ⪯, cf. [31] (Theorem 2.8). The Wigner–Yanase skew information is thus the maximal element among the Wigner–Yanase–Dyson skew informations with respect to the order relation ⪯.

### 7.3. The Monotonous Bridge

The family of metrics with representing functions
$$f_{\alpha }\left(t\right)=t^{\alpha }{\left(\frac{1+t}{2}\right)}^{1-2\alpha }\;t>0,$$
decrease monotonously (with respect to ⪯) from the largest monotone symmetric metric down to the Bures metric for α, increasing from 0 to 1. They correspond the the constant weight functions $h_{\alpha }\left(\lambda \right)=\alpha$ in Equation (Equation 7). However, the only regular metric in this bridge is the Bures metric (α=0). It is, however, possible to construct a variant bridge by choosing the weight functions
$$h_{p}\left(\lambda \right)=\left\{\begin{array}{ccc} 0,\; & \lambda & <1-p\\ p, & \lambda & \geq 1-p \end{array}\right.\;0\leq p\leq 1$$
in Equation (Equation 7) instead of the constant weight functions. It is non-trivial that these weight functions provide a monotonously decreasing bridge (with respect to ⪯) of monotone symmetric metrics between the smallest and the largest (monotone symmetric) metrics. The benefit of this variant bridge is that all the constituent metrics are regular, except for p=1.

## 8. Metric Adjusted Local Quantum Uncertainty

We consider a bipartite system $H=H_{1}⊗H_{2}$ of two finite dimensional Hilbert spaces.

**Definition** **10.***Let $f\in F_{\mathrm{op}}$ be regular and take a vector $\Lambda \in R^{d}.$ We define the* Metric Adjusted Local Quantum Uncertainty *(f-LQU) by setting*(11)$$U_{1}^{\Lambda ,f}\left(\rho_{12}\right)=\inf\left\{I_{\rho_{12}}^{f}\left(K_{1}⊗1_{2}\right)|K_{1}\;\mathrm{has}\;\mathrm{spectrum}\;\Lambda \right\},$$*where $\rho_{12}$ is a bipartite state, and $K_{1}$ is the partial trace of an observable K on H.*

The infimum in the above definition is thus taken over local observables $K_{1}⊗1_{2}\in B\left(H_{1}⊗H_{2}\right),$ such that $K_{1}$ is unitarily equivalent with the diagonal matrix diag(Λ).

**Remark** **5.**
*The metric adjusted LQU has been studied in the literature for specific choices of f.*

*If $f\left(x\right)=f_{WY}\left(x\right)={\left(\frac{1+\sqrt{x}}{2}\right)}^{2}$, then $U_{1}^{\Lambda ,f}$ coincides with the LQU introduced in [13] (Equation (Equation 2)).*

*If $f\left(x\right)=f_{SLD}\left(x\right)=\frac{1+x}{2},$ then $U_{1}^{\Lambda ,f}$ coincides with the Interferometric Power (IP) introduced in [14].*



**Proposition** **5.**
*For $f,g\in F_{op}^{\;r}$ with $\tilde{g}\leq \tilde{f},$ we have the inequality $U_{1}^{\Lambda ,f}\left(\rho_{12}\right)\leq U_{1}^{\Lambda ,g}\left(\rho_{12}\right).$ This implies that the Interferometric Power is the biggest among the Metric Adjusted LQU; see Proposition 3.*


**Proof.** Let ${\tilde{K}}_{1}$ be the local observable with spectrum Λ minimizing the metric adjusted skew information. Then,
$$U_{1}^{\Lambda ,f}\left(\rho_{12}\right)=I_{\rho_{12}}^{f}\left({\tilde{K}}_{1}⊗1_{2}\right)\geq I_{\rho_{12}}^{g}\left({\tilde{K}}_{1}⊗1_{2}\right)\geq U_{1}^{\Lambda ,g}\left(\rho_{12}\right),$$
where we used the inequality in (Equation 6). □

**Corollary** **2.**
*Let $g_{1}$ and $g_{2}$ be regular functions in $F_{op}$ and set $f={\tilde{g}}_{1}\wedge {\tilde{g}}_{2}$ with respect to the lattice structure in $F_{op}\;.$ Then, there is a regular function g in $F_{op},$ such that $\tilde{g}=f={\tilde{g}}_{1}\wedge {\tilde{g}}_{2}$ and*
$$\max\left\{U_{1}^{\Lambda ,g_{1}}\left(\rho_{12}\right),\;U_{1}^{\Lambda ,g_{2}}\left(\rho_{12}\right)\right\}\leq U_{1}^{\Lambda ,g}\left(\rho_{12}\right)$$
*for arbitrary $\rho_{12}\;.$*


**Proof.** The functions ${\tilde{g}}_{1}$ and ${\tilde{g}}_{2}$ are non-regular by Theorem 4. By Proposition 4, we thus obtain that the minorant *f* is also non-regular. Therefore, from the correspondence in Theorem 4, there is a (unique) regular function *g* in $F_{op}$ such that $\tilde{g}=f.$ The assertion then follows by Proposition 5. □

Following [10], we prove that the metric adjusted LQU is a measure of non-classical correlations, i.e., it meets the criteria which identify discord-like quantifiers; see [4].

**Theorem** **5.**
*If the state $\rho_{12}$ is classical-quantum in the sense of [32], then the metric adjusted LQU vanishes; that is, $U_{1}^{\Lambda ,f}\left(\rho_{12}\right)=0.$ Conversely, if the coordinates of *Λ* are mutually different (thus rendering the operator $K_{1}$ non-degenerate) and $U_{1}^{\Lambda }\left(\rho_{12}\right)=0,$ then $\rho_{12}$ is classical-quantum.*


**Proof.** We note that the metric adjusted skew information $I_{\rho_{12}}^{f}\left(A\right)$ for a faithful state $\rho_{12}$ is vanishing if and only if $\rho_{12}$ and *A* commute. If $\rho_{12}$ is classical-quantum, then
$$P_{1}\left(\rho_{12}\right)=\sum_{i}\left(P_{1,i}⊗1_{2}\right)\rho_{12}\left(P_{1,i}⊗1_{2}\right)=\rho_{12}$$
for some von Neumann measurement $P_{1}$ given by a resolution $\left(P_{1,i}\right)$ of the identity $1_{1}$ in terms of one-dimensional projections. We may choose $K_{1}$ diagonal with respect to this resolution, so $K_{1}⊗1_{2}$ and $\rho_{12}$ commute, and thus $U_{1}^{\Lambda ,f}\left(\rho_{12}\right)=0.$If, on the other hand, the Metric Adjusted Local Quantum Uncertainty $U_{1}^{\Lambda ,f}\left(\rho_{12}\right)=0,$ then there exists a local observable $K_{1}⊗1_{2}$ such that $[\rho_{12},K_{1}⊗1_{2}]=0.$ Then, by the spectral theorem
$$K_{1}=\sum_{i}\lambda_{i}P_{1,i}=\sum_{i}\lambda_{i}{|i\rangle }_{1}\langle i|$$
for a resolution $\left(P_{1,i}\right)$ of the identity $1_{1}$ in terms of one-dimensional projections, and since
$$\rho_{12}\left(K_{1}⊗1_{2}\right)=\left(K_{1}⊗1_{2}\right)\rho_{12},$$
we obtain, by multiplying with $P_{1,i}⊗1_{2}$ from the left and $P_{1,j}⊗1_{2}$ from the right, the identity
$$\lambda_{j}\left(P_{1,i}⊗1_{2}\right)\rho_{12}\left(P_{1,j}⊗1_{2}\right)=\lambda_{i}\left(P_{1,i}⊗1_{2}\right)\rho_{12}\left(P_{1,j}⊗1_{2}\right).$$If $K_{1}$ is non-degenerate, it thus follows that
$$\left(P_{1,i}⊗1_{2}\right)\rho_{12}\left(P_{1,j}⊗1_{2}\right)=0\;\mathrm{for}\;i\neq j.$$By summing overall *j* differently from i, we obtain
$$\left(P_{1,i}⊗1_{2}\right)\rho_{12}\left(\left(1_{1}-P_{1,i}\right)⊗1_{2}\right)=0,$$
thus
$$\left(P_{1,i}⊗1_{2}\right)\rho_{12}=\left(P_{1,i}⊗1_{2}\right)\rho_{12}\left(P_{1,i}⊗1_{2}\right),$$
so $P_{1,i}⊗1_{2}$ and $\rho_{12}$ commute. It follows that
$$P_{1}\left(\rho_{12}\right)=\sum_{i}\left(P_{1,i}⊗1_{2}\right)\rho_{12}\left(P_{1,i}⊗1_{2}\right)=\rho_{12}\;,$$
so $\rho_{12}$ is left invariant under the von Neumann measurement $P_{1}$ given by $(P_{1,i}).$ Therefore, $\rho_{12}$ is classical-quantum. □

Recall that Luo and Zhang [33] proved that a state $\rho_{12}$ is classical-quantum if and only if there is a resolution $\left(P_{1,i}\right)$ of the identity $1_{1}$ such that
$$\rho_{12}=\sum_{i}p_{i}P_{1,i}⊗\rho_{2,i}\;,$$
where $\rho_{2,i}$ is a state on $H_{2}$ and $p_{i}\geq 0$ for each i, and the sum $\sum_{i}p_{i}=1.$ By [30] (Lemma 3.1), the inequality
$$I_{\rho_{12}}^{f}\left(K_{1}⊗1_{2}\right)\geq I_{\rho_{1}}^{f}\left(K_{1}\right)$$
is valid for any local observable $K_{1}\;,$ where $\rho_{1}=\mathrm{Tr}\;{\;}_{2}\;\rho_{12}\;.$ Consequently, we obtain that
(12)$$U_{1}^{\Lambda ,f}\left(\rho_{12}\right)\geq \underset{K_{1}}{\inf}\;I_{\rho_{1}}^{f}\left(K_{1}\right)=\underset{\sigma_{1}}{\inf}\;I_{\sigma_{1}}^{f}\left(K_{1}\right),$$
where the infimum is taken over states $\sigma_{1}$ on $H_{1}$ that are unitarily equivalent with $\rho_{1}.$

**Theorem** **6.**
*The metric adjusted LQU is invariant under local unitary transformations.*


**Proof.** For the metric adjusted skew information and local unitary transformations, we have
$$\begin{array}{c} U_{1}^{\Lambda ,f}\left(\left(U_{1}⊗U_{2}\right)\rho_{12}{\left(U_{1}⊗U_{2}\right)}^{†}\right)=\underset{K_{1}}{\inf}I^{f}\left(\left(U_{1}⊗U_{2}\right)\rho_{12}{\left(U_{1}⊗U_{2}\right)}^{†},K_{1}⊗1_{2}\right)\\ =\underset{K_{1}}{\inf}I^{f}\left(\rho_{12},{\left(U_{1}⊗U_{2}\right)}^{†}\left(K_{1}⊗1_{2}\right)\left(U_{1}⊗U_{2}\right)\right)\\ =\underset{K_{1}}{\inf}I^{f}\left(\rho_{12},(U_{1}^{†}K_{1}U_{1}⊗1_{2}\right)=U_{1}^{\Delta ,f}\left(\rho_{12}\right), \end{array}$$
where we used the definition in (Equation 11). □

**Theorem** **7.**
*The metric adjusted LQU is contractive under completely positive trace-preserving maps on the non-measured subsystem.*


**Proof.** A completely positive trace preserving map $\Phi_{2}$ on system 2 is obtained as an amplification followed by a partial trace (Stinespring dilation); that is,
$$\left(1_{1}⊗\Phi_{2}\right)\rho_{12}=\frac{1}{d_{3}}\mathrm{Tr}\;{\;}_{3}\left(\left(1_{1}⊗U_{23}\right)\left(\rho_{12}⊗1_{3}\right){\left(1_{1}⊗U_{23}\right)}^{†}\right),$$
where $d_{3}$ is the dimension of the Hilbert space of the ancillary system 3. The metric adjusted skew information is additive under the aggregation of isolated systems; that is,
$$I^{f}\left(\rho ⊗\sigma ,A⊗1_{2}+1_{1}⊗B\right)=I^{f}\left(\rho ,A\right)+I^{f}\left(\sigma ,B\right)$$
and trivially $I_{\rho }^{f}\left(A+I\right)=I_{\rho }^{f}\left(A\right),$ where *I* is the identity operator [17]. Therefore,
$$\begin{array}{cc} U_{1}^{\Lambda ,f}\left(\rho_{12}\right) & =I^{f}\left(\rho_{12},{\tilde{K}}_{1}⊗1_{2}\right)=I^{f}\left(\rho_{12}⊗\frac{1}{d_{3}}1_{3},{\tilde{K}}_{1}⊗1_{23}+1_{12}⊗1_{3}\right)\\ & =I^{f}\left(\rho_{12}⊗\frac{1}{d_{3}}1_{3},{\tilde{K}}_{1}⊗1_{23}\right), \end{array}$$
where ${\tilde{K}}_{1}$ is the local observable minimizing the metric adjusted skew information. The metric adjusted skew information is invariant under unitary transformations and contractive under partial traces. Therefore,
$$\begin{array}{cc} U_{1}^{\Lambda ,f}\left(\rho_{12}\right) & =\;I^{f}\left(\left(1_{1}⊗U_{23}\right)(\rho_{12}⊗\frac{1}{d_{3}}1_{3}\right)\left(1_{1}⊗U_{23}^{†}\right),{\tilde{K}}_{1}⊗1_{23})\\ & \geq I^{f}\left(\mathrm{Tr}\;{\;}_{3}\left\{\left(1_{1}⊗U_{23}\right)\left(\rho_{12}⊗\frac{1}{d_{3}}1_{3}\right)\left(1_{1}⊗U_{23}^{†}\right)\right\},{\tilde{K}}_{1}⊗1_{2}\right)\\ & =I^{f}\left(\left(1_{1}⊗\Phi_{2}\right)\rho_{12},{\tilde{K}}_{1}⊗1_{2}\right)\\ & \geq U_{1}^{\Lambda ,f}\left(\left(1_{1}⊗\Phi_{2}\right)\rho_{12}\right), \end{array}$$
where we again used [30] (Lemma 3.1). □

**Theorem** **8.**
*The metric adjusted LQU reduces to an entanglement monotone for pure states.*


**Proof.** The metric adjusted *f*-LQU coincides with the standard variance on pure states; that is,
$$I_{\rho }^{f}\left(A\right)={\mathrm{Var}}_{\rho }\left(A\right)=\mathrm{Tr}\;\;\rho A^{2}-{\left(\mathrm{Tr}\;\;\rho A\right)}^{2}$$
whenever ρ is pure [17] (Theorem 3.8). However, in [13] it has been proven that the minimum local variance is an entanglement monotone for pure states. □

## 9. Conclusions

In this work, we built a unifying information-geometric framework to quantify quantum correlations in terms of metric adjusted skew information. We extended the physically meaningful definition of LQU to a more general class of information measures. Crucially, metric adjusted quantum correlation quantifiers enjoy, by construction, a set of desirable properties which make them robust information measures.

An important open question is whether information geometry methods may help characterize many-body quantum correlations. In general, the very concept of multipartite statistical dependence is not fully grasped in the quantum scenario. In particular, we do not have axiomatically consistent and operationally meaningful measures of genuine multipartite quantum discord. Unfortunately, the LQU and IP cannot be straightforwardly generalized to capture joint properties of more than two quantum particles. A promising starting point could be to translate into the entropic multipartite correlation measures developed in [34] into information-geometry language. We plan to investigate this issue in future studies.

## Figures and Tables

**Table 1 entropy-23-00263-t001:** Means and associated functions.

Name	*f*	$m_{f}$
arithmetic	$\frac{1+x}{2}$	$\frac{x+y}{2}$
WYD, β∈(0,1)	$\frac{x^{\beta }+x^{1-\beta }}{2}$	$\frac{x^{\beta }y^{1-\beta }+x^{1-\beta }y^{\beta }}{2}$
geometric	$\sqrt{x}$	$\sqrt{xy}$
harmonic	$\frac{2x}{x+1}$	$\frac{2}{x^{-1}+y^{-1}}$
logarithmic	$\frac{x-1}{\log x}$	$\frac{x-y}{\log x-\log y}$

**Table 2 entropy-23-00263-t002:** Examples of $f-\tilde{f}$ correspondence.

*f*	$\tilde{f}$
$\frac{1+x}{2}$	$\frac{2x}{x+1}$
$\frac{{\left(\sqrt{x}+1\right)}^{2}}{4}$	$\sqrt{x}$
$\beta \left(1-\beta \right)\frac{{\left(x-1\right)}^{2}}{\left(x^{\beta }-1\right)\left(x^{1-\beta }-1\right)}$	$\frac{x^{\beta }+x^{1-\beta }}{2}$

## Data Availability

Not applicable.

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
