# Peer review of "A Unified Approach to Local Quantum Uncertainty and Interferometric Power by Metric Adjusted Skew Information"

_entropy, 2021, doi:10.3390/e23030263_

Round 1
Reviewer 1 Report
The article "A unified approach to Local Quantum Uncertainty and
Interferometric Power by Metric Adjusted Skew Information" revisits the problem of discord quantification from the perspective of families of operator monotones. It is shown that two quantifiers for quantum discord whose origins are founded in the non-vanishing commutator between a density matrix and an observable, are indeed two special cases of a more general information measure. Furthermore, the authors demonstrate relevant mathematical properties for the entire family of generalized discord measures, such as unitary invariance, contraction under CP maps, and reduction to an entanglement measure for pure states.
The result is very interesting and reveals a deeper, rigorous connection between so far only vaguely related discord measures. The presentation of the results is quite clear, and the overview of basic steps that lead to the definition of the considered quantities of interest is very useful for readers with less background in the construction of these families.
Overall, I believe this result is a valid addition to the literature. I recommend publication of the article after the authors have considered the following specific remarks which may help to further improve the article (ordered from top to bottom in the manuscript)
- Line 103: Here the term "a quantum Fisher information" appears for the first time, but so far this has not been formally introduced. It would be good to explain what the authors consider a quantum Fisher information early on, since this term is widely used but has different meanings in different communities. In quantum information, for example, the term quantum Fisher information is reserved for the unique maximum of the classical Fisher information over all POVMs [Braunstein & Caves '94].
- Most symbols are carefully introduced in the manuscript before their usage. However I believe on page 6, the symbol Dn1 has not been defined before.
- Below Eq.(5): typo in "summery"
- line 167: typo: H2 should also be in mathcal form.
- line 169: typo "coincide"
- I wonder if it might be worth mentioning convexity of the metric adjusted LQU (if it can be inherited from I) as another desirable property of a discord measure
- As the authors mention, for the LQU and IP analytic expressions for qubit systems exist. Is this also possible for the generalized metric adjusted LQU?
- The discussion on the metric adjusted LQU reminds me of the results of Ref. [27] on generalized Fisher informations that are also upper bounded by the variance. There it was shown that the Braunstein-Caves quantum Fisher information is special in the sense that it is the largest in that family, which also makes it the strongest entanglement witness. I wonder whether an analogous statement can be made here, i.e., can one identify an extremal metric adjusted LQU that is in a certain sense the most powerful one as a discord measure?
Author Response
All the suggested corrections have been implemented and a general revision of the paper has been done.
About the conceptual remarks.
1) We do not expect that a measure of quantum correlation should be convex.
2) It's certainly natural to try to generalize the expressions for the qubit case that we know for the LQU and IP. But this does not appear straightforward and we plane to address this delicate problem in a future publication.
3) In the corrected version we stress that the IP is, indeed, maximal among the metric adjusted LQU.
Reviewer 2 Report
Local quantum uncertainty and interferometric power are important concepts in quantum information theory, it is interesting to study them in a unified manner employing the metric adjusted skew information. I think this manuscript is interesting. It is also well organized and written. Only some notations need to be modified and unified, which has been marked in the manuscript.

Author Response
We fully revised the paper to eliminate typos.
Reviewer 3 Report
Dear Editor,
It seems that the paper contains new results and it seems the proofs are okay. Roughly speaking, the authors presented some results on "local quantum uncertainty and interferometric power " that was discussed by
1. D. GIROLAMI , T. T UFARELLI AND G. A DESSO . Characterizing Nonclassical Correlations via Local Quantum Uncertainty. Phys.
Rev. Lett. 110, 240402 (2013).
2. D. G IROLAMI , A. M. S OUZA , V. G IOVANNETTI , T. T UFARELLI , J. G. F ILGUEIRAS , R. S. S ARTHOUR , D. O. S OARES -P INTO , I. S.
O LIVEIRA AND G. A DESSO . Quantum discord determines the interferometric power of quantum states Phys. Rev. Lett. 112,
210401 (2014).
Indeed, in [1,2] "local quantum uncertainty and interferometric power " as geometric quantifiers of quantum correlations.
Here, the authors examined their properties in a unified manner by means of the metric adjusted skew information defined by the third author.
The presentation is good.
I have recommend it as it is.
PS:
I recommend to check the paper's grammar once again. For example in the abtsract, it was written "by means of the the metric adjusted" and
one of the "the" should be removed.
Author Response
To improve the style of the paper and to eliminate typos a full revision of the paper has been implemented.